# Growth and Behavior of North American Microbes on *Phragmites australis* Leaves

**DOI:** 10.3390/microorganisms8050690

**Published:** 2020-05-08

**Authors:** Aaron E. DeVries, Kurt P. Kowalski, Wesley A. Bickford

**Affiliations:** USGS Great Lakes Science Center, Ann Arbor, MI 48105, USA; adevries75@gmail.com (A.E.D.); wbickford@usgs.gov (W.A.B.)

**Keywords:** germination, invasion, disease, barcode, microbiome, pathogen, endophyte

## Abstract

*Phragmites australis* subsp. *australis* is a cosmopolitan wetland grass that is invasive in many regions of the world, including North America, where it co-occurs with the closely related *Phragmites australis* subsp. *americanus*. Because the difference in invasive behavior is unlikely to be related to physiological differences, we hypothesize that interactions with unique members of their microbiomes may significantly affect the behavior of each subspecies. Therefore, we systematically inoculated both plant lineages with a diverse array of 162 fungal and bacterial isolates to determine which could (1) differentiate between *Phragmites* hosts, (2) infect leaves at various stages of development, or (3) obtain plant-based carbon saprophytically. We found that many of the microbes isolated from *Phragmites* leaves behave as saprophytes. Only 1% (two taxa) were determined to be strong pathogens, 12% (20 taxa) were weakly pathogenic, and the remaining 87% were nonpathogenic. None of the isolates clearly discriminated between host plant lineages, and the *Phragmites* cuticle was shown to be a strong nonspecific barrier to infection. These results largely agree with the broad body of literature on leaf-associated phyllosphere microbes in *Phragmites*.

## 1. Introduction

The common reed is a cosmopolitan wetland grass found on lakeshores, estuaries, and most wet areas where it can produce dense stands to as much as 4 m tall. Although numerous closely related plant lineages have been recognized at or below the species level, only the European *Phragmites australis* subsp. *australis* (Cav.) Trin ex. Stuedel is widely regarded as an invasive species due to its habit of replacing diverse wetland ecosystems with large monocultures in North America. At the same time, many European populations of reed have been observed to decline rapidly and disappear in just a few years in a phenomenon commonly known as reed die-back [1]. In contrast, the North American continent is known to host several distinct *Phragmites* lineages, including the native *P. australis* subsp. *americanus* Saltonstall, P.M. Peterson and Soreng, in addition to the introduced *P. australis* subsp. *australis* plants along the east coast starting around the mid 1800s [2] and in the Gulf of Mexico [3]. Both native and introduced lineages now share a strongly sympatric distribution pattern, grow in similar habitats, and almost identical anatomies that differ in only minor morphological characters [4], yet only the *P. australis* subsp. *australis* lineage is known to invade wetland communities aggressively.

Because it is difficult to explain the differences in their invasive growth potential in terms of physiology or their physical environment, the difference may instead depend on their relationships with the biological environment. Interactions between plants and their microbiomes are well-known to affect plant growth, and it has been repeatedly suggested that *Phragmites* is no exception [5,6]. Evidence for this hypothesis is mixed, however, because *Phragmites* plants in the Great Lakes region are known to host a diverse assemblage of endophytes in their leaves and the microbial populations are largely determined by site-specific conditions [7]. A more recent study (2020) examining the diversity of foliar fungal communities across a latitudinal gradient along the east coast of the United States further found that host plant lineage and biogeographic gradients strongly influence foliar fungal community structure and pathogen susceptibility, but these did not translate to differences in pathogen damage patterns between plant lineages [8]. *Phragmites* seeds have also been found to carry numerous fungal endophytes that affect germination rates and seedling survival [9] and are occasionally infected by specific seed parasites [10], though it is not clear how much they affect seedling recruitment in either plant lineage. Alternatively, a mutualistic endophytic fungus (*Stagonospora* sp.) has been shown to increase *P. australis* subsp. *australis* growth under laboratory conditions [11]. Below ground, fungal root endophytes provide tolerance to salt-stress for plants growing in mesohaline sites [12], while the roots of populations experiencing die-back in Europe have been attributed to direct [13] or indirect [14] microbial associations in anoxic soils. Similarly, observations of the *Phragmites* rhizosphere microbiome in North America have found distinctly different communities of archaea [15] and bacteria [16]. However, others have found that these differences are weak and heavily dependent on stand density and site conditions [17,18]. Differences in root gallic acid secretion has been proposed to affect the rhizosphere microbiome of *Phragmites* roots, [19], though the low concentration and short half-life of gallic acid in nonsterile soils does not clearly support this hypothesis [20].

Thus, while there are some detectable differences in the microbial communities between *Phragmites* lineages, it is not clear why these differences exist or how they might influence plant behavior. Leaf tissue chemistry may play a role in shaping the microbiome community above ground, as the North American native and introduced *Phragmites* subspecies are known to differ in their photosynthetic capacity and chlorophyll content [21]. Leaf tissue chemistry also affects the carbon sources available to the microbe community before and after leaf senescence. The microbes may, in turn, affect plant growth and behavior directly through pathogenic interactions or by producing plant hormones [22]. Below ground, most plants are known to benefit from mycorrhizal associations with specific fungi that are attracted to specific root secretions or the oxygen gradients released by *Phragmites* roots in anaerobic soils [23]. These interactions are also likely to be complicated further by numerous relationships with other components of the soil (e.g., pH, redox), which may enhance or suppress potential plant interactions. However, very little is known about which microbes perform any of these functions in *Phragmites*, much less whether or not they can (1) differentiate between plant subspecies, (2) infect the plant at various stages of development, or (3) behave as pathogens, endophytes, or saprophytes to obtain plant-derived carbon.

To address these issues, we directly assayed the responses of two *Phragmites* lineages (subsp. *americanus* and *australis*) against a panel of 162 microbes in culture, all of which had been isolated from *Phragmites* leaves collected from North America. Specifically, we measured infection potential, infection severity, and saprophytic potential. To put our experimental results into a broader context, we further compared our results to a list of *Phragmites*-associated microbes that been reported from a variety of published and unpublished data sets around the world. Finally, we classified the characteristics of each microbes (pathogenicity, distribution, function) to identify species that may be targets for future biocontrol work.

## 2. Methods

### 2.1. Isolation and Identification of Live Microbe Cultures Taken from Phragmites Tissues

Endophytic microbes from presumably diseased leaves of multiple lineages of *P. australis* were collected from multiple wetlands across the Atlantic Coast, Gulf of Mexico, and the lower Great Lakes region. As described [8], leaf punches were surface-sterilized 1 min 95% EtOH, 2 min 5% bleach, and 1 min 70% ethanol (EtOH). The leaf punches were then left on 1% plain agar (21 °C, 50% RH, 16:8 h light:dark), isolating pure cultures of all culturable microbes (bacteria and fungi) as they emerged using hyphal tip or single spore methods. Stock cultures were maintained on corn meal agar (CMA) at 4 °C.

For identification, 200–600 mg of tissue was scraped off the surface of CMA plates containing pure cultures and extracted with a Qiagen Plant Mini kit (Qiagen, Hilden, Germany). The volume of buffers AP1 and AW1 were increased 2–3× as needed to minimize DNA loss from to the presence of agar in the tissue scrapings. Final DNA concentrations were adjusted to 1 ng/μL with a Qubit 3.0 fluorometer. Fungi were identified by amplifying the Internal Transcribed Spacer (ITS) region and the Large Subunit (LSU) of the nuclear ribosomal RNA (rRNA) using primers ITS1-F [24], ITS4 [25], and LR0R [26] and LR16 [27], respectively. Polymerase Chain Reactions (PCR) were performed in 50 µL volumes containing 1 ng genomic DNA and GoTaq polymerase^®^ (See Appendix A for primer sequences, PCR master mixes, and conditions). Prokaryotes were similarly identified using 16S and cpn60 barcodes following the methods described by [28]. Single products were purified using QIAquick PCR cleanup kit^®^ before being submitted to the University of Michigan Sequencing Core facility (Ann Arbor, MI) for Sanger sequencing. Contigs were edited and assembled using BioEdit^®^ version 7.2 Sequence Alignment Editor [29] and resulting fungi sequences were submitted to the MycoBank Database (http://www.mycobank.org) [30] for preliminary identification. The top 50 results were screened using 95% length and 99.0% similarity thresholds and clustered with Weighted Pair Group Method with Arithmetic Mean (WPGMA) to identify monospecific clades. Prokaryotic sequences were identified using NCBI BLAST search, filtered using threshold scores ≥1300, and clustered with the “Taxonomy” browser to identify monophyletic clades. Partial identifications were accepted when lower taxonomic ranks could not be resolved to a single name. Barcoded fungi were further assigned a Species Hypothesis codes (SH) from the UNITE database [31,32] at the 1.5% threshold to facilitate comparisons among unnamed and/or poorly resolved species. The names of fungi and oomycetes were reviewed for nomenclatural corrections through July 2019 using MycoBank [30] for fungi and BacDive https://bacdive.dsmz.de/ for bacteria and archaea [33]. All barcode sequences were deposited in Genbank, accession numbers found in Appendix A.

### 2.2. Mature Leaf Assay

We grew clones of both *Phragmites* lineages by cutting wild-collected rhizomes into pieces containing 3 nodes each, followed by burial in 2-gallon pots with a peat-based potting soil mix (Sungro Professional Growers Mix, SS#8-F2). The pots were placed in standard greenhouse flats and watered to a constant depth of 5 cm. Plants were grown in 16/8 h at a constant 30 °C. Approximately 3.7 L of fertilizer containing 0.32% Miracle Gro^®^ (28-8-16 NPK), 0.16% Earth Juice Microblast^®^ (Mg 0.50%-B 0.02%-Co 0.005%-Fe 0.10%-Mn 0.10%-Mo 0.0005%-Zn 0.05%), and 0.02% Sequestrene 330^®^ iron supplement was added to each flat on a weekly basis.

We prepared cultures of uniform size for inoculation by placing 6.2 mm disks of #1 Whatman^®^ filter paper on CMA cultures containing a pure culture of each microbe in our inventory. When saturated with live cells up to 20 disks at a time were placed inside a folded sheet of Parafilm^®^ separated in a 2 cm grid and moistened with 5 µL of sterile molecular grade H_2_O. Fully loaded parafilm sheets were wrapped around the mature leaves in the upper 1/3 of 6–8-month-old plants to form a flat sleeve, with the disks on the adaxial surface, and sealed to within 1 mm of the edge of the leaf (see Appendix A). Replicates performed simultaneously were applied to separate leaves of the same clone, and up to five different clones were used as they matured. Abraded test leaves were prepared in parallel by rubbing the adaxial surface lengthwise with three strokes of sterile steel wool, which removed a thin glaucous coat on the epidermis without causing excessive tissue damage. All cultures were removed after 24 h, and symptoms of infection (described below) were recorded after 10 days. Each 2 × 2 cm section of inoculated leaf lamina was photographed and given a score to rate disease severity (0 = healthy tissue, 0.5 = areas of discolored tissue ≤ 6.2 mm diameter, and 1 = areas of discolored or necrotic tissue ≥ than 6.2 mm diameter). Scores were averaged using between three and seven replicates.

### 2.3. Phragmites Seed Collection

We collected mature panicles of *P. australis* subsp. *australis* and *P. australis* subsp. *americanus* from the Great Lakes region. *P. australis* subsp. *australis* panicles were collected in November 2016 from Indiana Dunes National Park in Porter, IN, USA (41.66°, −87.04°). *P. australis* subsp. *americanus* panicles were collected in October 2017 from Cedar Point National Wildlife Refuge east of Toledo, OH, USA (41.68°, −83.29°). Panicles were air-dried and cold stratified at 4 °C for three months. Dry panicles were threshed by rubbing them through 18 × 16 mesh window screen and then mixed with water containing 0.05% Tween-20 to allow the seeds to settle. Excess water and chaff were poured off, while the seeds were air-dried overnight and then stored desiccated at 4 °C. Due to the high variability of wild-collected seeds, we selected those that displayed ≥80% viability and were ≤3% contaminated by endophytic fungi. Plant subsp. identifications were confirmed following the Restriction Fragment Length Polymorphism (RFLP) method described by Saltonstall [34].

### 2.4. Seedling Leaf Assay

Sterile seeds were germinated by modifying a protocol described for maize [35]: seeds were washed in 95% EtOH 5 min to break dormancy, followed by 1% bleach 3 min, 3× washes ddH_2_O, and overnight storage in room temperature water. The following day, the seeds were washed again 70% EtOH 2 min, 1% bleach 10 min, and 3× washes with ddH_2_O before being plated to 1% agar containing Gamborgs B5 salts and vitamins (3.2 g/L). To facilitate synchronous germination, the plated seeds were kept dormant by storing them in the dark at 4 °C for 1–3 weeks and then moved to a Percival^®^ growth chamber (16/8 h, 30 °C/18 °C day/night cycles). Plates were oriented vertically so that emerging roots remained on the surface of the agar. The plates were inspected daily for up to 5 days to remove seeds with surviving endophytes, after which all remaining plants were found to be endophyte-free. To test seedling sterility, we barcoded week-old sterile seedlings next to positive controls and found a negative result for all seedlings (unpublished data).

To perform the bioassay, uniformly sized seedlings (5–10 mm coleoptile, 10–15 mm root) were selected and gently pressed onto the surface of 1% H_2_O agar containing a pure culture of each microbe. Control plates were established by pressing a set of seedlings on 1% H_2_O agar with no culture added. Between 4 and 14 sterile plants were used per replicate, depending on the number of contaminated seedlings that were discarded. The remaining seedlings were incubated in the growth chamber for up three weeks while periodically recording disease symptom development using a 0–4 Disease Index (D.I.) scale on leaves: 0 = dark green, 1 = pale green, 2 = yellow, 3 = brown spots present, 4 = dead. These were then used to calculate a weighted average, $\frac{\left(N_{0}\times 0\right)+\left(N_{1}\times 1\right)+\left(N_{2}\times 2\right)+\left(N_{3}\times 3\right)+\left(N_{4}\times 4\right)}{\left(N_{0}+N_{1}+N_{2}+N_{3}+N_{4}\right)}$, where *N*_0_–*N*_4_ are the total number of leaves in each category. Preliminary data indicated that this scale produced a highly linear curve during the infection phase, so standard 10-day D.I. values were calculated from linear regression through the origin for each replicate. Data from plants older than 16 days were excluded from these calculations, and in cases where the microbe killed all leaves, only the first two data points where D.I. = 4.0 were included. Similar calculations were also performed to estimate the growth rate of the seedlings, measured in leaves/day. Paired *t*-tests were used to compare treatments with controls and to compare treatments performed in parallel on both plant subspecies.

Because the roots were also visible in this assay, we also recorded incidental observations of hyphal growth patterns and found three distinct routes of infection: (1) infections that began at the root cap and was followed by a slow acropetal spread of hyphae up the root and into the lower stem and leaves, (2) infections that destroyed the first roots they encountered, but all subsequent roots produced by the plant remained healthy, and (3) infections that rapidly destroyed the leaves, but left behind a healthy but decapitated root system. For a full list of observations see DeVries et al. [36].

### 2.5. Saprophyte Assay

Circular punches (6.2 mm dia.) were collected from mature leaves on 6–8 months old stems of *P. australis* subsp. *americanus*, autoclaved for 15 min, and placed on 50 g of moist sterile sand in a 10 cm petri dish with 2 cm spacing. To inoculate, small blocks of CMA media (≤ 0.5 mm^2^) containing a fresh culture of each fungi were placed on the sand 1 mm from the edge of each leaf punch to distinguish agar-based growth from growth supported by the leaf-tissue. Bacterial cultures were grown in liquid lysogeny broth (LB) for 2 days before being 5 µL were transferred to the center of the leaf punch. Completed plates were sealed with parafilm, incubated in the growth chamber, and periodically monitored at under a dissecting microscope at 40× magnification for up to 50 days.

### 2.6. Literature-Based Microbiome Comparison

To determine what fraction of the *Phragmites* microbiome was included in our collection of microbes, we assembled a dataset containing a list of microbes known to have been identified in, on, or near *Phragmites* tissues. An initial set of recent papers was identified by searching public databases for *Phragmites australis* and its former name *Phragmites communis,* along with keywords including “reed”, “fungi”, “bacteria”, “archaea”, “oomycetes”, “microbe”, and “virus”. The citations in the initial set of papers were then traced iteratively backwards through time until no new citations could be located. Reports of individual microbes were accepted where the author had personally identified the microbe from infected *Phragmites* tissues and/or rhizosphere soil samples. Previous collections reused by subsequent papers were not included. All published barcode sequences were reidentified as described above for consistency. Only microbes completely identified to the species level or had distinct UNITE Species Hypothesis information were used for comparative analysis.

Global distribution patterns were recorded based on the current geopolitical boundaries, and GPS coordinates were estimated based on author descriptions. Due to the existence of multiple closely related plant lineages in the *Phragmites* genus [2,3,37,38], we chose to analyze two lineages in North American: *P. australis* subsp. *P. americanus* Saltonstall, P.M. Peterson and Soreng, and *P. australis* subsp. *P. australis* Trin ex. Steudel. The “Gulf” lineage was previously identified as interspecific hybrid, *Phragmites mauritianus* × *P. australis* [37] and was excluded from further analysis, while the “Delta” and “Greeny” lineages were classified *P. australis* subsp. *P. australis* as per the previous genetic analysis [3,38,39]. Reports from central and western North America host plants collected before 1910 were classified *P. australis* subsp. *americanus* because they predated the spread of invasive *P. australis* subsp. *australis* into that area [2]. European plants meanwhile are known to have distinct genetic populations centered around the Mediterranean basin and northern Europe [37]. In the absence of clear morphological characters, we distinguished these populations on a geographic basis as follows: reports from plants in Spain, Portugal, Italy, Egypt, and Morocco were classified as *P. australis* subsp. *P. altissimus*, while the remainder of European reports were classified as *P. australis* subsp. *australis.* Romanian plants were classified as “uncertain” due to the presence of hybrids in this area [37]. Other reports from Asia, Australia, Saudi Arabia and Africa were assumed to be from *P. australis* subsp. *australis* unless otherwise specified by the author. Area-proportionate Venn diagrams were generated with EulerAPE 3.0.0 [40]. Fungi and Oomycetes were further assigned to predicted ecological guilds using FUNGuild, a tool that estimates functional potential of fungal taxa based on observations in the literature [41]. Because many reports identified the precise plant tissue, we further divided each report as necessary to facilitate analysis between stems, leaves, leaf sheaths, inflorescences, seeds, rhizomes, and other below-ground organs. Reports taken from the subterranean tissues were more difficult to classify because few sources precisely identified the complex underground structures produced by *Phragmites*, which include buried stem bases, scale leaves, rhizomes, long gravitropic primary adventitious roots, and short secondary roots. Therefore, we recognized only three categories of reports from subterranean structures: rhizosphere soil, rhizomes, and a homogenized mixture of primary and secondary roots.

To estimate community composition in data obtained from literature reports, we filtered the data to make comparisons among like groups. To avoid sampling bias, our community composition estimates included only the papers that were culture dependent (98 of 119). Of those culture dependent studies, we selected only those which included at least 10 isolates. We then separated each study by the particular part of the plant producing the isolates and considered each a separate community. This resulted in 70 communities for comparison (47 Fungi, 13 Bacteria, and 10 Oomycetes). We explored differences in community composition using the Bray-Curtis similarity index. Communities were compared between continents, *Phragmites* lineages, and plant parts using Per-MANOVA.

## 3. Results

### 3.1. Microbe Identification

We identified 162 operational taxonomic units (OTUs) from our collection of isolates. The vast majority of isolates (85%) were fungal, of which 96% (133 isolates) were of the phylum *Ascomycota*, with the remaining from *Basidiomycota* (7 isolates). The most common fungal genera identified were *Stagonospora* (13 isolates), *Alternaria* (11 isolates), and *Paraphaeosphaeria* (6 isolates). Bacterial isolates were of the phyla *Actinobacteria* (7 isolates), *Firmicutes* (9 isolates), and *Proteobacteria* (6 isolates). See Appendix A and DeVries et al. [36] for full report.

### 3.2. Mature Leaf Assay

None of the 162 isolates tested induced lesions on healthy mature *Phragmites* leaves within the 10-day incubation period, yet if the leaf surface was lightly abraded before inoculation, 127 microbes (78%) could induce at least minor lesions (D.I. ≥ 0.1). Due to high sensitivity of this assay to environmental conditions, only 8 isolates were identified as potential pathogens (i.e., producing minor lesions and significantly different than controls; Figure 1A, Table 1), and 35 isolates failed to produce lesions. Comparisons of disease severity between lineages indicated that isolates were more likely to produce lesions in subsp. *australis* than subsp. *americanus* (*t* = −4.2613, *df* = 88, *p* ≤ 0.001), however the majority of isolates were not significantly different from the controls (Figure 1A).

### 3.3. Seedling Leaf Assay

To improve the reliability of our assay, we assessed disease symptoms of sterile seedlings raised in a controlled environment using a modified disease index scale (D.I.). During the experiment, the growth of control plants was repeatedly observed to slow down after three weeks and occasionally displayed the yellowed leaves with green veins characteristic of iron deficiency. Similar symptoms are also known from mature plants [42], indicating that this syndrome may affect all stages of the *Phragmites* life cycle. Based on the degree of yellowing observed, only D.I. values greater than 0.03 were considered to be symptoms caused by the microbes. Pathogenic species were further defined as having D.I. scores ≥ 2.0 due to the irreversible color changes and necrosis that occur above this value. Using this threshold, 13 fungal isolates could be considered pathogenic on *P. australis* subsp. *australis,* though only two of them were considered significantly different from controls at *P_0_* ≤ 0.05 (see Table 1). Parallel assays using *P. australis* subsp. *americanus* were limited by seed supplies, so only a subset of 52 potential pathogens could be tested. These tests revealed a strong correlation between D.I. values on *P. australis* subsp. *americanus* and *P. australis* subsp. *australis* (Figure 1B) and identified very nearly the same set of pathogenic microbes (Table 1). None of the pair-wise comparisons were found to differ from each other at the *P_0_* ≤ 0.05 threshold, indicating that the tested microbes did not distinguish between the two host plant lineages.

### 3.4. Saprophyte Assay

To identify microbes capable of saprophytic growth on dead plant tissue, we used all 162 cultures to inoculate autoclaved *Phragmites australis* leaves and kept them moist for up to thirty days. Using this method, 73 fungi species were found to produce conidia or micro-sclerotia, including 11 species that produced gelatinous conidial masses. Another 23 fungal species only grew as sterile hyphae, and five failed to grow at all within thirty days. These data suggest that at least 74% of the fungi were capable of completing their life cycle on dead *Phragmites* tissue under the conditions used here, while the remainder likely have more specialized requirements for spore production or grow as obligate endophytes.

### 3.5. Literature-Based Microbiome Comparison

In order to estimate what proportion of the *Phragmites* microbiome is represented by our collection of fungi and bacteria, we took advantage of the voluminous literature on the subject to compile a list of *Phragmites*-associated microbes that had been identified by other authors. Our survey included 93 papers and 2 alternative sources and documented 10,514 distinct reports of microbes found on or inside *Phragmites* tissues, as well as those collected from rhizosphere soil samples (see Appendix A and DeVries et al. [36]). To facilitate accurate comparisons with our data, we focused on a subset of 2829 reports were the taxa were identified to the species level, including 372 partially identified fungi that were distinguished using their Species Hypothesis codes (UNITE database [31,32]). Together these reports represented 27% of the database.

### 3.6. Taxonomic Distribution of the Phragmites Microbiome

After performing the nomenclatural corrections as described in the methods, this subset of reports was found to include 1547 distinct taxonomic units at the species level, corresponding to 605 fungi, 43 oomycetes, 82 bacteria, 10 archaea, and 7 viruses. The fungi were strongly represented by *Ascomycota* (480 taxa), with smaller numbers of *Basidiomycota* (74 taxa), *Mucoromycota* (3 taxa), *Mortierellomycota* (3 taxa), *Glomeromycota* (7 taxa), and *Monoblepharomycota* (1 taxon). The oomycetes were more difficult to characterize due to the large number of partially identified reports, but those that were identified were dominated by a single genus *Pythium* spp. (33 taxa) and included *Saprolegnia* spp. (3 taxa) and 1 taxon each for *Aphanomyces*, *Brevilegnia, Phytopthora, Phytophythium, Plectospira, Dictyuchus, and Achlya* sp. Bacteria were strongly represented by *Proteobacteria* with (43 taxa) but included the phylum Firmicutes (18 taxa), Actinobacteria (16 taxa), Bacteroidetes (4 taxa), and a single Spirochaete report. The taxonomic distribution of Archaea could not be determined due to the small number of identified taxa, but those that were identified included several methane and ammonia-oxidizing genera including *Methanosarcina, Methanococcus, Methanofollis Nitrosocomicus,* and *Nitrosarchaeum* sp. Four genera of ssRNA viruses were identified as well, including *Luteovirus*, *Potyvirus*, *Poleovirus*, and *Tospovirus* sp.

### 3.7. Geographic Distribution of the Published Phragmites Microbiome

Globally, the locations where the *Phragmites* microbiome had been sampled by other authors closely approximated the known distribution of the plant, which is abundant in Europe and North America, with lesser amounts in Africa, the Middle East, Asia, and South America. Over 99.9% of all identified reports were mapped to plants in the Northern Hemisphere (Figure 2), which were roughly equally distributed between Europe (312 taxa) and North America (268 taxa), with smaller numbers from Asia (64 taxa). The Southern Hemisphere *Phragmites* microbiome was represented by just 6 taxa reported from Sub-Saharan Africa and Australia. Taxonomically, the fungi were well-represented in Europe, where they had been recorded from more than half of the individual countries. In North America, the fungi were distributed into two distinct sampling areas, one centered around the state of Michigan and spreading west, and the second formed a discontinuous band that closely followed the east coast from Maine, around Florida and into the Gulf of Mexico. In Asia, most recent reports of fungi were concentrated in Hong Kong, though there were many old reports from Japan. The majority of bacteria were reported from Europe (54 taxa), with reduced numbers for North America (13 taxa) and Asia (22 taxa). The geographic distribution of reported oomycetes was even more limited, as European reports were concentrated around Germany and Poland (11 taxa), while north American reports were clustered around Michigan, New York, and New Jersey (13 taxa). Fully identified archaea (30 taxa) have thus far only been reported from Spain [43], though an unspecified number of *Phragmites*-archaea interactions have been studied in Maryland [15]. The distribution of *Phragmites* viruses displayed no clear geographic pattern.

The literature-based data also clearly indicated that the *Phragmites* microbiome is organized into distinct regional floras. As shown in Figure 3A, the flora of each continent had very little overlap, with only 34 taxa reported from two or more continents. The shared taxa were also found to be enriched in agriculturally significant species (some beneficial, some pathogenic; see Figure 3B), suggesting that they reflect an anthropogenic distribution pattern. The continental pattern was also consistent between the major groups of organisms, as fungal communities were further found to be different among continents (*r*^2^ = 0.135, *p* = 0.001) and lineages (*r*^2^ = 0.103, *p* = 0.051). However, some plant lineages were geographically isolated (*P. australis* subsp. *americanus* and Gulf lineage), so their differences are more likely due to geography alone. Bacterial communities showed few differences regionally (*p* = 0.23) or among lineages (*p* = 0.66). Communities of oomycetes also differed by continent (*r*^2^ = 0.319, *p* = 0.034), however given the disparate number of studies between the two continents (Europe *n* = 6 and North America *n* = 4), the communities had unequal dispersion and may not differ significantly.

In North America, the literature described 304 distinct species of fungi and bacteria that occurred on one or both plant lineages. However, these displayed little evidence of host specificity, as *P. australis* subsp. *americanus* and *P. australis* subsp. *australis* shared between 63% and 45% of their individual microbiomes, respectively (Figure 3C). Fungal communities within North America also did not differ among lineages (*p* = 0.51) or plant parts (*p* = 0.40). We were unable to make similar calculations for other taxonomic groups due to the small number of reports that were recorded from *P. australis* subsp. *americanus*.

### 3.8. Potential Tissue-Specific Patterns

Due to the often-ambiguous descriptions of plant organs collected below ground, we relied instead on descriptions of tissue sizes and the apparatus used to process them. We found that 224 subterranean taxa were originally collected from a mixture of the primary and secondary roots, while 30 taxa were clearly collected from rhizome tissues and zero were recorded from buried stem bases or scale leaves. Our preliminary analysis further revealed a substantial overlap between root and rhizome microbiome taxa (Figure 3B), so these were combined into a single “root-associated” microbiome for all subsequent analysis. In addition, several authors have noted that the microbial flora changes abruptly on *Phragmites* stems located above and below water [44,45], but these reports were not included in our analysis because they were all based on dead plant tissues. After accounting for these filters and including taxa that were associated with stems and leaves, we found 529 globally reported microbial taxa potentially associated with *Phragmites*. However, contributions of these microbes to the plant overall health, whether positive (beneficial) or negative (pathogenic), needs to be established. Broadly, no single microbe was reported from all *Phragmites* tissues, though a *Stagonospora* sp. (SH1576715.08FU) from North America was notable for being reported from all but root and seed tissues. Nor were any fungi found to be closely related to the beneficial *Epichloë* and *Balansia* spp. endophytes found in other grasses [46]. Instead, the literature revealed only a handful of probable obligate endophytes from the North American flora, included *Engyodontium ablum, Neovossia moliniae*, and two partially identified members of the Ustilaginaceae. When the microbiome of living plant tissues sorted into phyllosphere and rhizosphere populations, we found that these two broad categories of *Phragmites* anatomy displayed clearly distinct microbiomes with a moderate degree of overlap (See Figure 4A). However, the high ratio of phyllosphere/rhizosphere taxa is inconsistent with the expected populations in each environment, which may reflect the difficulty of collecting below-ground samples. Both phyllosphere and rhizosphere communities were dominated by fungi, though the root-related flora recorded 3–4× more oomycetes, bacteria, and archaea than their respective phyllosphere communities.

Above ground, we further divided the *Phragmites* microbiome into leaf blade (264 taxa), leaf sheath (79 taxa), stems (89 taxa), seeds (24 taxa), and inflorescence branches (3 taxa) communities. Excluding the seed and inflorescence communities, there was a moderate amount of overlap between the leaf and sheath microbiomes and between leaf and stem microbiomes, while the stem and sheath communities were anomalously distinct despite being in direct physical contact with each other (Figure 4C). Rather than being a sign of tissue specificity, this pattern may be related to seasonal plant growth patterns, as the young stem is fully enclosed by the sheath in the spring and only becomes exposed to late-season microbes as the leaf sheath elongates and senesces in the fall. Accurate comparisons with the seed and inflorescence microbiomes could not be performed due to the small number of reports for each. Below ground, the microbiome could only be grouped by roots and rhizomes due to the lack of reports for other tissue types (Figure 4B). We found that the majority of reported microbes were identified on root tissues, though this may affected by sampling bias as these were recorded by only one North American source [7].

### 3.9. Predicted Trophic Feeding Behavior

Due to the potential to misclassify the host plant tissue as dead or alive using bibliographic data, we used the FUNGuild database [41] to predict the trophic mode of each fungal taxon in the global bibliographic data set and then compared them to our experimentally determined behaviors. For this analysis, we selected fungi reported from both living and dead *Phragmites* tissues, as well as those reported from rhizosphere soil samples. After being classified by FUNGuild, we excluded the least likely predictions (classified as “possible”) and excluded taxa recognized only at the family level or higher, which produced a list of 602 fungi. When these were clustered by trophic modes, the majority were predicted to be pathogens and saprotrophs, with much smaller proportions of predicted endophytes and various mixed life cycles (Figure 5). Proportionally, the global ratio of all predicted Pathogen/Endophyte/Saprophytes was roughly 6:1:9. Similar but less precise ratios (see DeVries et al. [36]) were also detected on each continent, potentially indicating a large-scale community organizing principle. Communities grouped by trophic mode differed among continents (*r*^2^ = 0.115, *p* = 0.004) indicating that along with differences in the community structure, the functions may also differ slightly. The functional potential of the fungi associated with subsp. *australis* also differed by continent (*r*^2^ = 0.152, *p* = 0.003) indicating that fungal communities may be functioning differently in the invaded and native range of subsp. *australis*. However, when relative abundances of each trophic mode were calculated, we determined that North American and European subsp. *australis* plants did not differ in pathogen abundance (19.7% and 17.4%, respectively; *p* = 0.96).

Following identification of species, our collection of 162 microbes was found to include 110 distinct taxa, representing 94 fungi and 16 bacteria (85% and 15% of the collection, respectively). Based on the total reported number of taxa that occur in the North American microbiome, we further estimate that our collection includes 50% of fungi taxa and 65% of the bacteria taxa that may occur on *Phragmites*. We also classified the trophic modes for all 162 microbes in our collection using our bioassay data and found that the fungi had Pathogen/Endophyte/Saprophyte trophic mode ratio of 1:1:12, which differed from their predicted ratio of 3:1:12. Even allowing for inaccuracies in the predicted data, this suggests that our collection of microbes under-represents the proportion of North American *Phragmites* pathogens by 3×. Furthermore, our collection was found to represents less than 1.3% of the European microbiome and 1.8% of the Asian *Phragmites* microbiome, respectively.

## 4. Discussion

Our experimental results indicate that none of our leaf-collected isolates could clearly discriminate between *Phragmites* lineages, suggesting that our sampled set of microbes may not have a strong role in explaining the invasive growth of *Phragmites australis* subsp. *australis* in North America. Although individual microbes did not show host-specificity for one lineage over the other in either assay, the mature leaf assay, but not the seedling assay, indicated that pathogen damage was stronger on leaves of subsp. *australis* on average. Our data are largely consistent with Allen et al. [8] who found that both plant lineages hosted a diverse community of fungi in leaves, yet produced little or no difference in pathogen damage between hosts. However, if our samples underestimate pathogens in the environment, it is possible that some specificity could exist in the environment that benefits the invasive subsp. *australis*. Additionally, all of the microbes in our collection were derived from phyllosphere populations, and thus the role of rhizosphere organisms in regulating *Phragmites* growth remains untested.

In the rhizosphere, others have found differentiation in composition between *Phragmites* lineages [16], but functional differences have been very subtle [47]. For instance, previous research has identified that oomycete pathogen communities differed between subsp. *australis* and *americanus* in the rhizosphere [6] but also that soil oomycetes did not show complete host-specificity [48]. Additionally, virulence of oomycete pathogens differed between lineage of *Phragmites* as well as other native plants [48]. These results are consistent with our findings in the phyllosphere, suggesting the rhizosphere may behave similarly. However, additional research is needed to characterize both compositional and functional differences of microbes in the rhizosphere and evaluate how those microbes influence plant productivity.

Although seedling plants were reliably inoculated by our procedures, this result could not be replicated in mature leaves without first damaging the cuticle. This strongly suggests that the cuticle is a powerful and non-specific barrier to infection. A similar but weaker barrier also appeared to exist on seedling leaves, where observational data revealed only ten microbes that could infect young leaves directly, whereas most appeared to infect the roots first and then slowly spread up into the leaf bases (based on multiple observations of roots and lower stems). The same acropetal pattern was also observed in 6-month old *Phragmites* stems whose roots were inoculated with a *Stagonospora* sp. fungus [11], raising the possibility that the lower stem may have higher numbers root endophytes than more distal portions. Such acropetal growth might also contribute to the large proportion of microbes reported from both phyllosphere and rhizosphere populations (Figure 4A).

The majority of microbes isolated from *Phragmites* tissues in our study were found to be saprophytes, with a much smaller subset found to be pathogenic and not lineage-specific. Surprisingly, there is little evidence in our samples that the exotic subsp. *australis* is less impacted by pathogens than the native subsp. *americanus* (Enemy-Release Hypothesis [49]), as both lineages were impacted similarly by pathogens as seedlings and subsp. *australis* was more strongly impacted on average by pathogens on mature leaves. These results are consistent with other studies of the phyllosphere [8] and root endophytes [17] of *Phragmites australis*, suggesting that the phyllosphere microbes may reflect broader landscape assemblages of microbes (i.e., those present on all plants in a region) and either influence the decomposition of *Phragmites* biomass or have no well-defined functions.

The large number of experimentally verified saprophytes in our collection of microbes is inconsistent with the source of these microbes, which were all isolated from living plant tissues and might be classified as endophytes on that basis. The disparity might be explained by the rapid colonization of dead or wounded plant tissue, where the microbe might easily be protected from common surface sterilization techniques by the remaining epidermal tissue. Alternatively, many of the saprophytes in our collection may instead grow as latent endophytes that are temporarily suppressed by living plant tissues but grow as saprophytes after the leaf senesces. Such a hypothesis might also explain why our collection of microbes had fewer pathogens than expected, as only two were clearly pathogenic and another 20 displayed weak pathogenic tendencies, though additional testing is required.

Global observations of microbes associated with *Phragmites* were consistent with our experimental findings and indicated that most observed microbes in *Phragmites* populations were likely saprotrophs, with a smaller subset as pathogens and endophytes. However, the reliability of FUNGuild to predict pathogenicity should be scrutinized given the results of our disease assays. The literature review also indicated that community structure of the microbiome within and surrounding *Phragmites* plants is strongly regionally regulated. Floras of fungi (the most common and well distributed group in our dataset) differed among continents. Further analysis indicated that the functional potential of those fungi found on *Phragmites* subsp. *australis* differed between continents. However, this difference was primarily driven by floras from Asia, and the relative abundance of pathogens did not differ between invaded and native ranges. This is consistent with other studies that have found no consistent differences in functional potential between *Phragmites* lineages [8,17,18] and may further support the understanding that enemy release is not a prominent factor in explaining the invasive character of subsp. *australis* in North America.

### Implications for Reed Biocontrol

The invasive growth of *P. australis* subsp. *australis* in North America and elsewhere in the world has stimulated much interest in developing potential control measures, ranging from grazing, plowing, burning, and herbicide treatments. While these methods are at least partially effective, Kowalski et al. [5] have advocated the development of a new biological control tool in part because endophytic relationships between plant and microbe are known to enhance (or suppress) plant growth. However, our results indicate that achieving this goal by applying *Phragmites* foliar pathogens is likely to be difficult because all mature leaves of this wetland grass are protected by a thick cuticle. This cuticle limits leaf-borne microbes from entering the plant, causing systemic infection, and affecting plant growth. New surface-applied microbe-based treatments of *Phragmites* will have to overcome this defensive feature of *Phragmites* leaves to ensure that the treatments are entering the plants.

Although our samples did not reveal any pathogenic microbes that clearly differentiate *Phragmites australis* subsp. *australis* from *Phragmites australis* subsp. *americanus*, we identified and tested only 27% of the fungal taxa globally known to associate with *Phragmites* [31,32]. Our samples were collected from visibly diseased leaves in North America along the Atlantic coast, the Gulf coast, and the southern Great Lakes, so they likely represent the suite of foliar microbes present in the central to eastern United States. However, the majority of reported microbes were untested. Additional study of the broader pool of microbes may reveal other taxa that are pathogenic or inhibit growth of *Phragmites*.

Finally, advances in the genetic sequencing technology and bioinformatics will continue to increase our capability to detect and identify new microbial taxa. The disease index described in this paper will be a useful tool to evaluate the pathogenicity of newly identified taxa and help prioritize development of those that appear to impact *Phragmites* the most. The most pathogenic microbes could be considered for development into a microbe-based treatment for invasive populations of *Phragmites*. Similarly, the index could be used by the agricultural sector to evaluate potential uses of or threats by newly identified microbes.

## Figures and Tables

**Figure 1 microorganisms-08-00690-f001:**
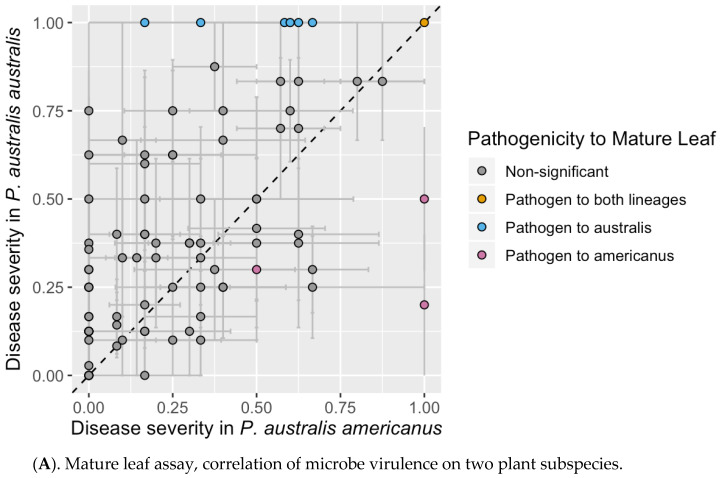
Pathogen assays comparing *P. australis* subsp. australis and *P. australis* subsp. *americanus.* Points indicate average values of each isolate’s disease index value from (**A**) abraded mature leaf assay (Disease Index (D.I.) Scale = 0–1) and (**B**) seedling leaf assay (D.I. Scale = 0–4). Dashed 1:1 lines are shown for comparisons. Colors indicate significant difference from controls at *P_0_* ≤ 0.05 with a Bonferroni correction for multiple comparisons.

**Figure 2 microorganisms-08-00690-f002:**
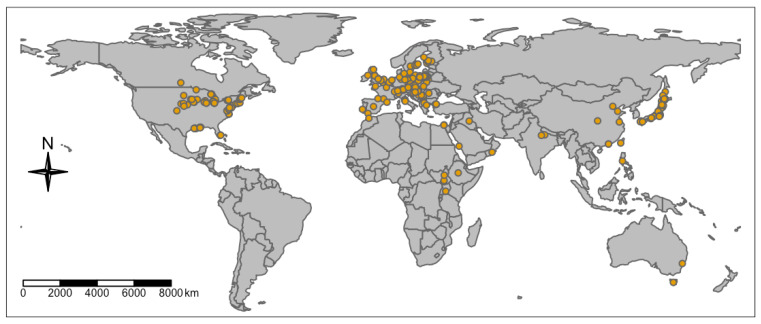
Locations where the *Phragmites australis* microbiome has been studied, based on a survey of literature reports and 37 new collections (this paper). See Appendix A for references and DeVries et al. [36] for GPS coordinates.

**Figure 3 microorganisms-08-00690-f003:**
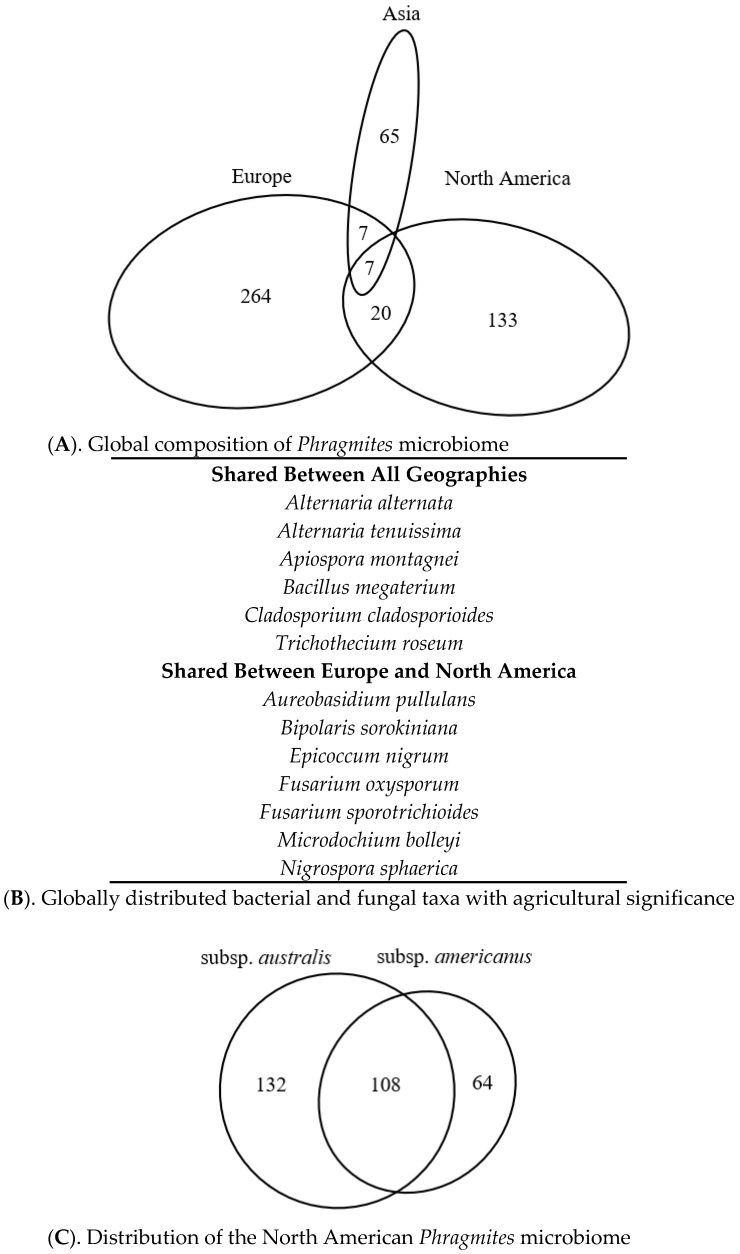
Geographic distribution of the *Phragmites* microbiome, based on all species reported worldwide. (**A**). Northern Hemisphere microbiome grouped by continent. (**B**). List of known agricultural pathogens and endophytes that occur in the 34 shared species. (**C**). North American microbiome grouped by plant host subsp. Area of ellipses is proportionate to the number of reports. See Appendix A for references and DeVries et al. [36] for species names.

**Figure 4 microorganisms-08-00690-f004:**
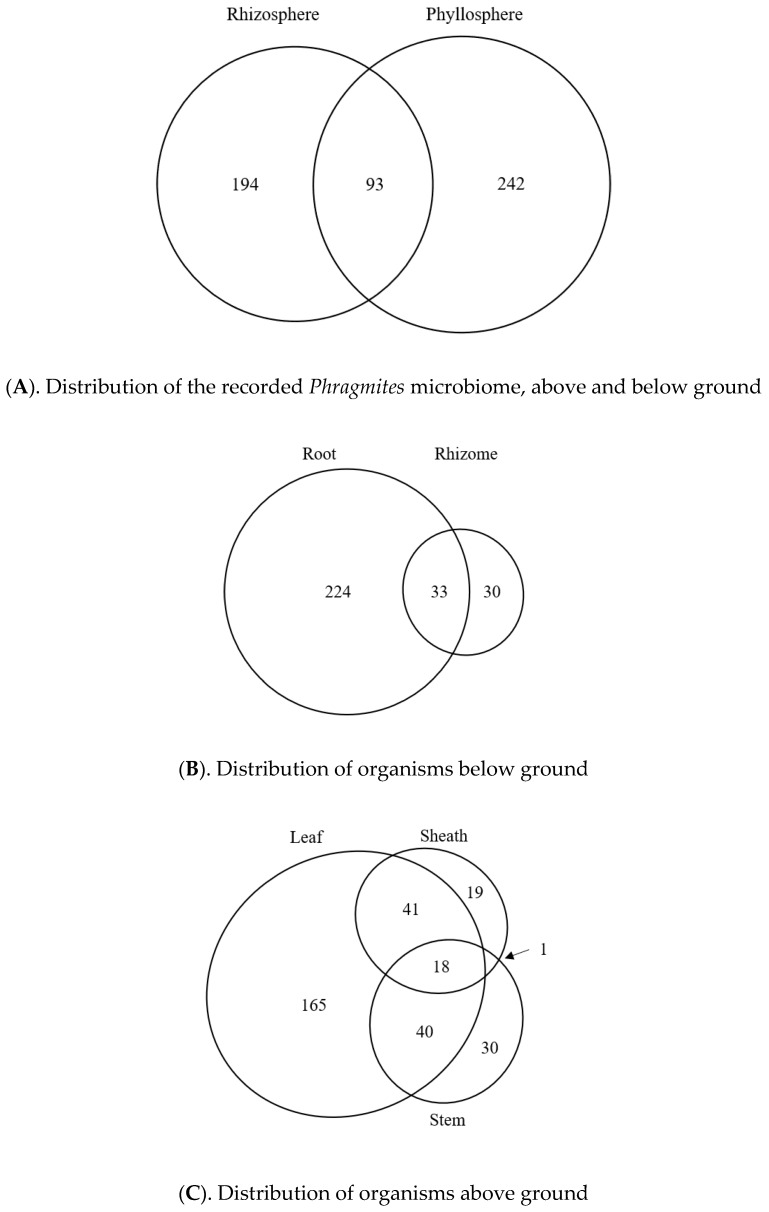
Tissue-specific microbiomes of *Phragmites australis,* based on literature reports from North America, Europe and Asia. (**A**). Comparison of above ground and below ground microbiomes. (**B**). Comparison of root and rhizome microbiomes. (**C**). Comparison of stem, leaf, and sheath microbiomes. Fungi reported from seeds and inflorescence branches are not shown due to the small number of reports. Area of ellipses is proportionate to the number of reports. See Appendix A for references and DeVries et al. [36] for species names.

**Figure 5 microorganisms-08-00690-f005:**
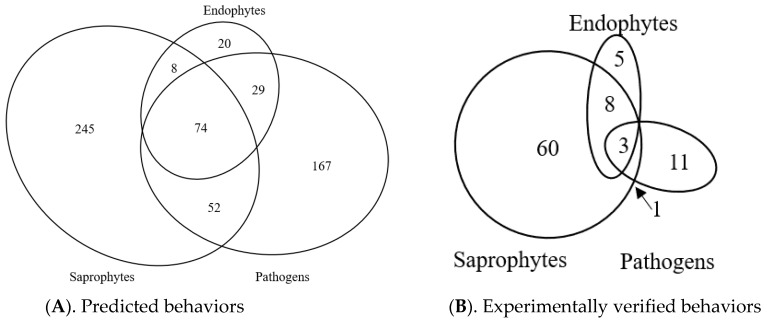
Classification of *Phragmites*-fungi interactions. (**A**). Predicted trophic modes, including all reported species worldwide. (**B**). Experimentally supported classification of 88 North American species. Figures are scaled to show the relative number of uniquely identified microbe species for each. See Appendix A for references and DeVries et al. [36] for species names.

**Table 1 microorganisms-08-00690-t001:** Confirmed *Phragmites australis* pathogens, as determined by mature leaf and seedling leaf assays. Mature leaf pathogen determinations made using Disease Index (D.I.) values. 0–0.2: Nonpathogen, 0.2–0.5: Weak Pathogen, 0.5–0.8: Pathogen, 0.8–1.0: Strong Pathogen. Seedling disease severity roughly translates to 0–2.0: Nonpathogens, 2.1–4.0: Pathogen, 4.1+: Strong Pathogen. Bold values were significantly different from controls at *P_0_* ≤ 0.05. *Previously identified as interspecific hybrid, *Phragmites mauritianus* × *P. australis* [37].

OTU	Subsp. of Original Host from Field Collection	Original Host Tissue	Putative Status of Original Host Tissue	Identification	Species Hypothesis (UNITE)	Pathogen Determination Mature Leaf		Disease Severity Seedlings	
						subsp. *australis*	subsp. *americanus*	subsp. *australis* (mean D.I.)	subsp. *americanus* (mean D.I.)
USGS23	*americanus*	Seed	Diseased	*Curvularia inaequalis*	SH1890305.08FU	**Strong Pathogen**	Pathogen	6.6	5.8
USGS32	*australis*	Leaf	Diseased	*Bipolaris sorokiniana*	SH1526399.08FU	**Strong Pathogen**	Weak Pathogen	3.7	4.1
USGS30	*australis*	Leaf	Diseased	*Pleosporaceae* sp.	SH1547057.08FU	**Strong Pathogen**	**Strong Pathogen**	3.5	2.2
LSU0038	*americanus*	*Leaf*	*Diseased*	*Pleosporales* sp.	SH1525096.08FU	**Strong Pathogen**	Pathogen	**3.2**	3.1
USGS15	*australis*	*Seed*	*Diseased*	*Septoriella hubertusii*	SH2176229.08FU	Weak Pathogen	**Strong Pathogen**	**2.9**	**2.5**
USGS17	*americanus*	*Seed*	*Diseased*	*Alternaria* sp.	SH1526398.08FU	**Strong Pathogen**	Nonpathogen	2.6	2.9
LSU0361	Gulf lineage *	*Leaf*	*Diseased*	*Colletotrichum sp.*	SH1543705.08FU	**Strong Pathogen**	Pathogen	2.6	1.6
LSU0719	*australis*	*Leaf*	*Diseased*	*Epicoccum sorghinum*	SH1547058.08FU	Pathogen	**Strong Pathogen**	1.2	1.6
LSU0107	*australis*	*Leaf*	*Diseased*	*Stagonospora neglecta*	SH1525143.08FU	**Strong Pathogen**	Pathogen	0.2	n/d
LSU0172	*australis*	*Leaf*	*Diseased*	*Stagonospora neglecta*	SH1525143.08FU	Strong Pathogen	Strong Pathogen	2.9	2.9
USGS07	*americanus*	*Stem*	*Diseased*	*Gibberella fujikuroi*	SH1610159.08FU	Strong pathogen	Pathogen	1.7	2.8
LSU1247	Gulf lineage *	*Leaf*	*Healthy*	Unknown fungus	n/a	Pathogen	Weak Pathogen	3.6	3.8
USGS34	*australis*	*Leaf*	*Diseased*	*Curvularia* sp.	SH1526408.08FU	Pathogen	Weak Pathogen	3.1	2.3
USGS42	*australis*	*Leaf*	*Diseased*	*Fusarium sporotrichioides*	SH2456045.08FU	Pathogen	Nonpathogen	0.3	2.8
LSU0677	Gulf lineage *	*Leaf*	*Diseased*	*Microdochium* sp.	SH1555458.08FU	Weak Pathogen	Weak Pathogen	4.6	3.7
USGS02	*australis*	*Leaf*	*Diseased*	*Stagonospora neglecta*	SH1525143.08FU	Weak Pathogen	Pathogen	2.2	1.3
LSU0240	*americanus*	*Leaf*	*Diseased*	*Gibberella fujikuroi*	SH1610159.08FU	Nonpathogen	Not tested	2.3	2.5

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
