# Peer review of "Growth and Behavior of North American Microbes on *Phragmites australis* Leaves"

_microorganisms, 2020, doi:10.3390/microorganisms8050690_

Round 1

Reviewer 1 Report

Very nice and systematic study and generates much new and important information.

Author Response

Thank you for the feedback. We hope this paper is an important contribution to the literature.

Reviewer 2 Report

The authors should be commended for both an extensive survey of microbes isolated from diseased Phragmites from North America, and the ability of these microbes to cause disease in mature Phragmites leaves, and Phragmites seedlings. This work is complemented by a thorough literature survey of the worldwide culturable microbiome of Phragmites. The aim of this work was to ascertain whether their was any differential susceptibility to any pathogens among the collection between the invasive introduced australis sub-species and the native americanus sub-species. The result that there was little differentiation is important for bio-control strategies.

With respect to the results, I would like to have the description of the microbe collection described earlier in the results, so the reader can have an idea of what organisms are being used in the assays. Was there any knowledge as to the sub-species the microbes were isolated from? Some clarity as to the endophyte free status of the seedlings used for the seedling assay should be made, they were assumed endophyte free due to no outgrowth? Are the microbes in the collection available under an MTA, have they been stored in any culture collection?

In figure 1 the colours for the categories other than pathogen to both lineages are very hard for a (red/green) colour blind reader such as the reviewer to differentiate.

In table 1 the range of the disease severity scores for the seedling assay should be explained, in a similar manner to those for the mature leaf assay. What was the control for the seedling assay, a set of seedlings pressed onto water agar with no culture added? Both columns in the seedling assay are labelled subsp australis, I assume one would be americanus.

In figure 3b it would be good to know if each of the named taxa are within the 20 taxa shared across all geographies, or in the two pools of 7 shared between 2 geographies. Is there a statistical measure of the enrichment?

The comparison between the authors collection and the taxa identified through the literature survey should be given a little more emphasis. The authors collection is from the phyllosphere, is the overlap with North American literature identified taxa much stronger with literature taxa isolated from the above ground parts of the plants. The low level of overlap of the authors collection with literature taxa from other geographies supports the regional flora hypothesis.

Are there any references supporting the accuracy of the predicted tropic data, with experimentally assessed data such as that presented by the authors. This has implications for the conclusion that the collection has an under-representation of pathogens. Was there any prediction of the pathogen causing the disease in the leaves from which the collection was originally isolated?

Are there culture independent molecular descriptions (16S ITS) of taxa from Phragmites that could be compared to the authors collection?

A few typos noticed:

line 157 mm rather the m for coleoptiles

line 321 in Hong Kong

line 482 phyllosphere not phyllospere

Overall the paper describes a substantial body of work that forms a culture based assessment of phyllosphere microbes from North American Phragmites that will be very useful for comparisons to culture-independent methods that will appear, and provides information to the bio-control community about how this method could be applied to invasive Phragmites in North America.

Author Response

The authors should be commended for both an extensive survey of microbes isolated from diseased Phragmites from North America, and the ability of these microbes to cause disease in mature Phragmites leaves, and Phragmites seedlings. This work is complemented by a thorough literature survey of the worldwide culturable microbiome of Phragmites. The aim of this work was to ascertain whether their was any differential susceptibility to any pathogens among the collection between the invasive introduced australis sub-species and the native americanus sub-species. The result that there was little differentiation is important for bio-control strategies.

Response: Thank you for the thoughtful review of our work.

With respect to the results, I would like to have the description of the microbe collection described earlier in the results, so the reader can have an idea of what organisms are being used in the assays.

Response: We apologize for the oversight, but one of the supplementary files was not uploaded appropriately. The identification of all microbes is listed in Supplemental table 1. We also added a brief paragraph at the start of the results to describe the identified microbes.

Was there any knowledge as to the sub-species the microbes were isolated from?

Response: Yes. This information is available in Appendix S1. We make reference in the text in lines XX. We also added the original host subspecies to Table 1 so it is clear in the description of pathogens.

Some clarity as to the endophyte free status of the seedlings used for the seedling assay should be made, they were assumed endophyte free due to no outgrowth?

Response: To test their sterility, we barcoded week-old sterile seedlings next to positive controls and found a negative result for all seedlings (unpublished data). Text added to methods.

Are the microbes in the collection available under an MTA, have they been stored in any culture collection?

Response: No. we have not deposited them in a formal culture collection. However, we are keeping them in our own personal collection. Therefore, we have the ability to send them out in the future, but we have not gone through the process yet.

In figure 1 the colours for the categories other than pathogen to both lineages are very hard for a (red/green) colour blind reader such as the reviewer to differentiate.

Response: We did use a colorblind friendly palette, but we see now that the chosen colors could be difficult. Fig 1 updated.

In table 1 the range of the disease severity scores for the seedling assay should be explained, in a similar manner to those for the mature leaf assay.

Response: Added text to the caption: “Seedling disease severity roughly translates to 0-2: Non-pathogens, 2.1-4.0: Pathogen, 4.1+: Strong Pathogen.”

What was the control for the seedling assay, a set of seedlings pressed onto water agar with no culture added?

Response: That is exactly what it was. We did it about 50 times. Text added to methods, “Control plates were established by pressing a set of seedlings on 1% H20 agar with no culture added.” 

Both columns in the seedling assay are labelled subsp australis, I assume one would be americanus.

Response: Yes. Thank you for catching that. Fixed.

In figure 3b it would be good to know if each of the named taxa are within the 20 taxa shared across all geographies, or in the two pools of 7 shared between 2 geographies. Is there a statistical measure of the enrichment?

Response: We created two headings, “Shared between all geographies” and “Shared between Europe and North America” to designate which were found in each group.

The comparison between the authors collection and the taxa identified through the literature survey should be given a little more emphasis. The authors collection is from the phyllosphere, is the overlap with North American literature identified taxa much stronger with literature taxa isolated from the above ground parts of the plants. The low level of overlap of the authors collection with literature taxa from other geographies supports the regional flora hypothesis.

Response: The overlap with the above ground parts is extremely strong. All LSU cultures came from leaves, as did most of the USGS cultures (this data now found in Supplemental table S1).

Are there any references supporting the accuracy of the predicted tropic data, with experimentally assessed data such as that presented by the authors. This has implications for the conclusion that the collection has an under-representation of pathogens.

Response: We do not know of other studies that have actually tested the accuracy of predicted trophic modes from databases like FUNguild. Our results suggest that these databases are not reliable as predictors of pathogenicity. However, they may be more reliable for other trophic modes which have a phylogenetic signature (mycorrhizae, some saprotrophs). That said, we use FUNguild to predict trophic modes of our literature based microbiome results. This analysis should therefore be qualified with a statement of the accuracy for the North American samples we actually tested. We added a clarification on this in the discussion. Added text: “However, the reliability of FUNguild to predict pathogenicity should be questioned given the results of our disease assays”.

Was there any prediction of the pathogen causing the disease in the leaves from which the collection was originally isolated?

Response: Yes, tissues sample’s status was recorded in the field or lab prior to isolating microbes. This data is now included in Supplemental S1. In addition, it was added to Table 1.

Are there culture independent molecular descriptions (16S ITS) of taxa from Phragmites that could be compared to the authors collection?

Response: Yes, we did include culture independent data from studies whose data was public. For analyses of diversity and community composition, those studies were removed to remove bias associated with much larger samples.

A few typos noticed:

line 157 mm rather the m for coleoptiles

Response: Revised.

line 321 in Hong Kong

Response: Revised.

line 482 phyllosphere not phyllospere

Response: Revised.

Overall the paper describes a substantial body of work that forms a culture based assessment of phyllosphere microbes from North American Phragmites that will be very useful for comparisons to culture-independent methods that will appear, and provides information to the bio-control community about how this method could be applied to invasive Phragmites in North America.

Response: Thank you for the thoughtful review of our manuscript. Your suggestions and clarifications have made the work better.